# Physical, Mechanical, and Biological Properties of PMMA-Based Composite Bone Cement Containing Silver-Doped Bioactive and Antibacterial Glass Particles with Different Particles Sizes

**DOI:** 10.3390/ma16134499

**Published:** 2023-06-21

**Authors:** Marta Miola, Giovanni Lucchetta, Enrica Verné

**Affiliations:** 1Applied Science and Technology Department, Politecnico di Torino, Corso Duca degli Abruzzi 24, 10129 Turin, Italy; marta.miola@polito.it; 2Department of Industrial Engineering, University of Padova, Via Venezia 1, 35131 Padua, Italy; giovanni.lucchetta@unipd.it

**Keywords:** bioactive glasses, bone cement, PMMA, antibacterial, silver

## Abstract

In the present work, antibacterial composite bone cement was designed by introducing a bioactive and antibacterial glass into a commercial formulation. The effect of glass particles’ addition on the curing parameters of the polymeric matrix was evaluated; moreover, the influence of the glass particle size on the glass dispersion, compressive and bending strength, bioactivity, and antibacterial effect was estimated. The results evidence a delay in the polymerization kinetics of the composite cement, which nevertheless complies with the requirements of the ISO standard. Morphological characterization provides evidence of good dispersion of the glass in the polymeric matrix and its exposition on the cement surface. The different glass grain sizes do not affect the composites’ bioactivity and compressive strength, while a slight reduction in bending strength was observed for samples containing glass powders with greater dimensions. The size of the glass particles also appears to have an effect on the antibacterial properties, since the composites containing larger glass particles do not produce an inhibition halo towards the S. aureus strain. The obtained results demonstrate that, by carefully tailoring the glass amount and size, a multifunctional device for artificial joint fixing, temporary prostheses, or spinal surgery can be obtained.

## 1. Introduction

Surgical site infections (SSIs) are one of the worst adverse events in surgery, since they are associated with longer post-operative hospital stays, additional surgical procedures, treatment in intensive care units, and often fatal outcomes. The incidence of SSIs in developed countries averages around 2–3%. This level of risk is significantly higher in developing countries, where SSI rates range from 1.2 to 23.6 per 100 surgical procedures [1]. In the past 70 years, antibiotics have been crucial in the fight against infectious diseases caused by bacteria and other microbes, with them being an important reason for the increase in average life expectancy in the 20th century. But just a few years after the mass production of antibiotics, microbes began to resist them, and antibiotics were at risk of becoming useless. New resistance mechanisms are emerging and spreading globally, threatening our ability to treat common infectious diseases. Antimicrobial resistance (AMR) is the cause of an increase in health care costs due to prolonged illness, additional tests, and the use of more expensive drugs, and it is associated with an increased risk of compromised clinical outcomes, disability, and death [2]. Today, the control and prevention of infectious diseases are one of the topics of the 2030 Agenda for Sustainable Development adopted by all United Nations Member States in 2015 (Sustainable Development Goal 3: Ensure healthy lives and promote well-being for all at all ages) [3].

Polymethylmethacrylate (PMMA)-based bone cement is commonly used for artificial joint fixing and as a filler for large bone defects in orthopedic surgery [4] due to its fast primary fixation to the bone. However, the interface between bone cement and bone has been recognized as a weak-link zone, both mechanically and biologically [5], as well as for its predisposition to bacterial contamination and, in turn, the risk of being a preferential site of surgical infection. Bacterial adhesion on the cement surface can be prevented by favoring their fast bone-bonding ability and with this purpose, the addition of bioactive fillers, like hydroxyapatite, bioactive glasses, and glass-ceramics, represents a common approach [6,7,8,9,10,11]. Another strategy to counteract bacterial contamination is to load the bone cement with antibiotics, like gentamycin, vancomycin, or others [9,12,13,14,15], but these formulations are mainly indicated for revision surgery due to some relevant drawbacks of antibiotic-loaded bone cement [16,17], including the risk of developing antibiotic-resistant strains, which strongly limits their use in prophylaxis.

Aiming to prevent the increase in multiresistant bacteria and to reduce the incidence of periprosthetic infections, some antibiotic-free antibacterial bone cement formulations have been investigated in the literature, for example loading bone cement with silver nanoparticles [9,18,19,20,21] or adding a variety of antibacterial additives together with bioactive fillers [9,22,23], but the antibacterial properties of these formulations, as well as bone cement handling and its mechanical properties still need optimization [24].

A completely different approach was patented by Vernè and co-workers [25], who developed a composite bone cement consisting of a PMMA matrix loaded with silica-based silver-doped bioactive glass particles. Bioactive glasses (BGs) are known worldwide as optimal materials for the realization of bone substitutes as well as coatings on metallic devices and can be treated to enrich their surfaces with a variety of active ions [26,27].

The use of silver-doped bioactive glasses as an additional phase in PMMA-based bone cement has previously been proven, by the authors of the present paper, to be safe and effective in various types of commercial bone cement, with them having different compositions and viscosities [28,29,30,31,32]. For each of the investigated composite formulations, deep studies have been carried out in order to assess the most effective glass synthesis methods, as well as the proper amount and grain size of the antibacterial and bioactive glass particles to modulate the bioactive and antibacterial ability of the implant. On the basis of these previous studies, the ion exchange process has been recognized as a good and versatile technique for the synthesis of bioactive and antibacterial glasses, since it allows for the introduction of a controlled and reproducible silver amount in the glass network without affecting its bioactivity by tailoring the process parameters: temperature, time concentration, and pH of the solution [29]. The composite bone cement described in the above-mentioned previous studies has been suggested as a very promising formulation due to the following proven advantages:Its bioactive and antibacterial properties are imparted by a unique inorganic phase (i.e., there is no need to embed too many additional phases into the bone cement).The silver ions release can be tailored and assured for a prolonged time, if necessary, through the correct design of the glass composition.The mechanical compressive strength of the composite bone cement is unaffected in comparison with the plain bone cement and is still in agreement with the ISO 5833 standards.The biocompatibility of plain bone cement is maintained.The antimicrobial effect has been demonstrated towards the most common bacterial and fungal strains.Bioactive and antibacterial glasses possess intrinsic radio opacity, so the use of traditional radio-opaque additives, in addition to the glass, is no longer needed.

These advantages enable, in case of future clinical use, an important reduction in implant septic loosening incidence and a significant decrease in antibiotic treatments due to periprosthetic infections.

For a better assessment of the potential of this innovative approach to prevent infection at surgical sites, the method for the preparation of composite bone cement containing bioactive and antibacterial glass particles has been further optimized and investigated in the present paper. Since it has already been reported by the authors that different viscosities of bone cement do not seem to influence the composites’ handling and the glass distribution in the PMMA matrix [30], in the present paper, high-viscosity commercial bone cement was used as a polymeric matrix, and the study was focused on the effect of glass particle size on the composite cement mechanical properties, in particular bending strength, without altering the bioactive and antibacterial behavior induced by the dispersed glass particles, as well as the setting properties of pristine cement.

## 2. Materials and Methods

### 2.1. Glass and Composite Cement Preparation

The glass synthesized in this work is a silica-based BG with the composition (mol%) of 48% SiO_2_, 18% Na_2_O, 30% CaO, 3% P_2_O_5_, 0.43% B_2_O_3_, and 0.57% Al_2_O_3_, named SBA2 from now on. The SBA2 BG was prepared by the traditional melt and quenching route, as reported in the literature [29,30]. Briefly, reagent-grade precursors were melted for 1 h in a platinum crucible at 1450 °C; the melt was then quenched in water obtaining a frit, which was ball milled and sieved down to two different grain sizes, i.e., <20 μm and between 20–45 µm. The two ranges were chosen on the basis of previous experience [30]. In particular, the glass powders with a grain size < 20 μm were prepared using a mortar (Pulverisette 2, Fritsch) which simulates the manual grinding process with the aim of avoiding the formation of fine particles (<5 μm). In fact, it was demonstrated that the presence of very fine particle fractions can produce a negative effect on the mechanical properties, as reported in [30]. Aiming to give antibacterial properties to the BG, an ion-exchange silver doping process was performed, soaking SBA2 powders into an aqueous solution of AgNO_3_ (0.03 M) according to previous works [24,29]. Successively, the silver-doped BG powders (Ag-SBA2) were dried in a heater at 60 °C.

The composite bone cement (CBCs) was prepared by mixing 10%wt of Ag-SBA2 powders with the commercial high-viscosity bone cement Cemex^®^ Isoplastic, kindly provided by Tecres S.p.A. Via A. Doria, 6—37066 Sommacampagna (VR) Italy. Plain commercial Cemex^®^ Isoplastic was used as the control. This bone cement is composed of a solid phase (spherical pre-polymerized PMMA, BaSO_4_ as a radio-opaque agent, and benzoyl peroxide as an initiator) and a liquid containing the monomer (methylmethacrylate—MMA), N,N-dimethyl-toluidine as an activator and hydroquinone as an inhibitor. Composite bone cement was prepared by mechanically mixing the Ag-SBA2 powders with the cement solid phase for 1 h until good dispersion of glass in the pre-polymerized PMMA was achieved. Afterward, the liquid phase was blended with the mixed powders, using the solid/liquid phase 3:1 ratio. The blend was stirred for about 1–1.5 min and, when it no longer stuck to the gloves, it was introduced into polished aluminum molds with dimensions useful for further characterization. The composite formulation was optimized considering the polymeric matrix viscosity and evaluating the glass particle size distribution.

From now on, the samples prepared with the BG powders with a grain size < 20 µm and in the range of 20–45 µm will be named C20 and C45, respectively.

### 2.2. Morphological and Compositional Characterization

The powders and both the plain and composite bone cement were subjected to morphological and compositional analyses by means of scanning electron microscopy (SEM—FEI, QUANTA INSPECT 200) and energy dispersion spectrometry (eDs—EDAX PV 9900). The analyses on bulk samples were performed on the samples surface, cross-section, and fracture surface after the bending test in order to estimate the glass exposition to the cement surface, its distribution into the polymeric matrix, and its role in the fracture mechanism. For the cross-section evaluation, the CBCs samples were cut using a cut-off machine (Accutom 5—Struers). Morphological analysis was also carried out after in vitro bioactivity tests.

### 2.3. Setting Time

The curing parameters of the composite bone cement were determined following the ISO 5833 (2002) standard [31] for the evaluation of the exothermic temperature changes occurring in PMMA bone cement during the setting process and to determine the four phases of polymerization: mixing, dough, working and setting time. As reported in the literature [32], the mixing time is the time needed to fully incorporate the powder and liquid. During the dough time, the cement achieves a suitable viscosity for handling (it can be handled without sticking to gloves). The working time is the period during which the cement can be handled, and the prosthesis can be inserted (it also results in an increase in viscosity and the generation of heat). The setting time is the time point measured from the beginning of mixing until the time at which the cement raises its temperature to a value that is halfway between the room and maximum temperature; during the setting time, the cement cures completely, and the temperature reaches its peak.

The SBA2 + PMMA powders were mechanically mixed and blended with MMA as described above and, after the mixing time, placed in a mold with a thermocouple (as required by the ISO standard). The temperature increase was recorded every 30 s. From the recorded data, the maximum temperature (T_max_), the time of T_max_, the setting temperature (T_set_), the setting time, and the dough time were extrapolated. Since the effects on the setting time are mainly due to the amount of BG particles introduced into the composite, which can slow down the polymerization reaction or partially prevent it, different glass amounts (10, 15, and 20 wt%) were investigated using C45 samples. Plain bone cement was used as a control.

### 2.4. Mechanical Characterization

Compression and bending tests [31] were performed on composite samples containing 10%wt of SBA2 powders with a grain size < 20 µm and in the range 20–45 µm (C20 and C45), and plain bone cement was used as a control with the procedures reported below.

The two ranges were chosen on the basis of previous experience [30], considering that the presence of very fine particle fractions can produce a negative effect on the mechanical properties [30,32], while larger glass powders size can induce a slight improvement in mechanical strength, as evidenced by the authors in a previous study with different polymeric matrices [30].

#### 2.4.1. Compression Test

Six samples for each formulation (C20, C45, and plain bone cement) were prepared as reported for the setting time test and shaped in a cylindrical mold for the compression test (6 mm diameter and 12 mm height). After curing, each sample was extracted from the mold and polished with SiC abrasive paper in order to assure plain and parallel surfaces.

The test was performed following ISO-5833 (2002) Annex-E “Determination of compressive strength of polymerized cement” [31] using a Sintec 10/D at a cross-head speed of 20 mm/min. All the obtained data are provided as means and standard deviations. All of the results were analyzed by Student’s *t*-test: *p* < 0.05 was considered significant.

#### 2.4.2. Bending Test

The same formulations were used to prepare six bars (75 × 10 × 3.3 mm) of C20, C45, and plain bone cement for the bending test (in accordance with ISO-5833-2002, annex F). The samples were cast in a rectangular mold and, after curing, polished with abrasive SiC paper (600) to reach the final dimensions. The four-point bending test was carried out following ISO-5833 (2002) annex F: “Determination of bending modulus and bending strength of polymerized cement” [32] and both the bending modulus (E_flex_) and the bending strength (σ_flex_) were determined. As for the compression test, all the obtained data are provided as means and standard deviations. All of the results were analyzed by Student’s *t*-test: *p* < 0.05 was considered significant. Moreover, a morphological analysis of the fracture surfaces after the bending test was carried out to estimate the influence of glass particles’ introduction on the mechanical properties.

### 2.5. Bioactivity

The bioactivity of the composite cement (i.e., the ability to induce the precipitation of hydroxyapatite in physiological conditions) was investigated by dipping samples in an acellular simulated body fluid (SBF, Kokubo [33]) for up to 28 days. Cylindrical samples of 5 mm-thick and 10 mm-diameter C20 and C45 samples with 10% of AgSBA2 and samples of plain bone cement as the control were soaked in 25 mL of SBF and maintained at 37 °C; the solution was refreshed every 2 days to simulate the renewal of body fluids, and the pH was monitored. At the end of the soaking time, the samples were subjected to SEM-EDS analyses. The tests were performed in triplicate.

### 2.6. Antibacterial Properties

The antimicrobial properties of the composite bone cement containing 10%wt of AgSBA2 glass powders both <20 μm and between 20–45 μm (C20 and C45) were investigated by means of the inhibition halo test, in accordance with the National Committee for Clinical Laboratory Standards (NCCLS [34]) using a Staphylococcus aureus strain (ATCC 29213), as reported in [28]. The samples were placed in contact with Mueller–Hinton agar, previously uniformly covered with bacteria, and incubated overnight at 35 °C to allow for bacterial growth. At the end of incubation, the formation of an inhibition zone was observed and measured. All products for this analysis have been purchased from BD-Becton Dickinson, Milano, Italia.

## 3. Results and Discussion

### 3.1. Morphological and Compositional Characterization

Figure 1 reports the morphology of the SBA2 particles < 20 μm (Figure 1a–c) and between 20–45 μm (Figure 1d–f); as expected, the glass particles have an irregular shape due to the milling process, and it is estimable that a glass sieved < 20 μm does not present fine particles < 5 μm. Figure 2 shows the morphology of the Cemex^®^ Isoplastic powders, in secondary (Figure 2a) and in backscattered electrons (Figure 2b) to better identify the presence of BaSO_4_. The images revealed pre-polymerized polymeric spheres with a variable diameter between about 10 and 80 μm together with BaSO_4_ agglomerates. Figure 2c the EDS analysis of the white spot showed the presence of barium sulfate, used as a radio-opacifier agent in the commercial formulation.

Figure 3 reports the morphology of Cemex^®^ Isoplastic after hardening, both the sample cross-section (Figure 3a) and the surface (Figure 3b). As can be observed, the sample shows a homogeneous morphology, with polymeric spheres surrounded by BaSO_4_ small particles, embedded after setting into the polymeric matrix. The composite bone cement, containing glass particles of different sizes (C20 and C45), presented similar homogeneity. Figure 4 reports, as an example, the morphology of C20 after hardening, both the sample surface (Figure 4a,b) and the cross-section (Figure 4c,d). Also, this sample shows a homogeneous morphology, with polymeric spheres surrounded by both glass and BaSO_4_ particles (see arrows) well embedded after setting into the polymeric matrix.

### 3.2. Setting Time

Figure 5 shows the temperature trend of each composite cement formulation, obtained from the setting time test, while Table 1 reports the mean value of setting time (t_set_) and temperature (T_set_) and the highest measured polymerization temperature (T_max_) and time (t_max_). As previously mentioned, in order to evaluate the effect of the inorganic phase on the curing parameters, for this test, the glass amount was increased up to 20 wt%.

The highest polymerization temperatures of the composites do not differ from the T_max_ of the Cemex^®^ Isoplastic; this is an expected result as the maximum temperature is linked to the amount of monomer, which causes an exothermic reaction through polymerization, and the MMA amount is equal for all the compositions. Instead, the time required to reach the maximum temperature increases by increasing the amount of glass in the composite. Concerning T_set_ and t_set_, the same trend was observed: there is not a significant difference among the T_set_ reached, but also, in this case, the setting time increases by increasing the glass amount, confirming a delay in polymerization kinetics, as also observed by other authors [35,36,37]. However, the obtained parameters satisfy the ISO requirements (from 3 to 15 min for t_set_ and a maximum temperature of 90 °C).

### 3.3. Mechanical Characterization

#### 3.3.1. Compression Test

Figure 6 shows the results of compressive strength evaluation according to ISO 5833-2002 [31]. Figure 6a reports the obtained average values of compressive strength, while Figure 6b shows as an example the typical stress–strain curve obtained for each sample. The compressive strength of all composites satisfies the ISO standard requirements (70 MPa); moreover, any significant difference was evidenced among the different formulations (Student’s *t*-test). Then, from a compression point of view, the size of the introduced glass powders does not seem to alter the compressive strength of the cement.

#### 3.3.2. Bending Test

To better investigate the mechanical behavior of the composites, and most of all to verify the influence of the glass introduction and the different grain sizes, the bending strength was estimated according to the ISO-5833 standard [31]. Figure 7 reports the bending strength and the bending modulus. As can be noticed, the commercial cement and the composite containing glass of <20 μm reached the bending strength imposed by ISO standards (50 MPa) (Figure 7a), while the composite containing glass with a grain size between 20 and 45 μm shows a bending strength slightly lower than the required one, even if the differences among the various formulation are not significant on the basis of the Student’s t-test. This behavior could be ascribed to the different grain sizes of the introduced glass. A small glass grain size and the elimination of the fine fraction allow us to obtain a bending strength comparable to the strength of commercial cement. Instead, as reported in the literature [37], the increase in the filler particle size reduces the bending strength. As reported for other PMMA-based materials [37], the presence of large particles in the polymeric matrix can increase the stress concentration at the interface between the PMMA matrix and glass. Moreover, the interaction between the polymer and glass is poor; there is only mechanical interlocking, which can decrease due to the shrinkage of the matrix during cooling.

Concerning the bending modulus, all the obtained values are above the limit imposed by the ISO standard (1800 MPa) (Figure 7b).

Figure 8 shows the micrographs of the fracture surfaces after the bending test. All the samples evidenced a distributed microporosity, connected to the manual mixing procedure, with a pore size between about 50–200 μm. The composite bone cement (Figure 8c–f) also showed BaSO_4_ particles and bioactive glass particles (highlighted by the circles in the insets) well dispersed in the polymeric matrix. The removal of the finest particle fraction allowed for avoiding the formation of agglomerates, which can act as critical defects and thus affect the bending strength of the composite, as observed in [30]. Therefore, the combination of an appropriate amount of glass with an optimized size can lead to the formation of composite cement with the same mechanical properties as commercial cement.

### 3.4. Bioactivity

The results for the in vitro bioactivity after 14 days of the SBF immersion test are reported in Figure 9. All types of composite cement showed the precipitation of a carpet of a needle-like submicrometric crystalline phase rich in Ca and P, organized with the typical globular morphology of in-vitro-grown hydroxyapatite (Figure 9c,g), which nucleated both on the cement surface and inside the pores (Figure 9a,e). It is worth mentioning that the nucleation of HAp takes place not only on the exposed glass particles but even on the PMMA surface (Figure 9a,b,e,f) through a glass-induced biomimetic effect. Any particular differences were noticed between the C20 and C45 samples, demonstrating the good dispersion and exposition of bioactive glass particles on cement surfaces for both compositions.

FESEM-EDS analysis after 28 days of SBF immersion (Figure 10) confirmed the bioactive behavior of the composites, with them showing a very thick and well-developed layer of hydroxyapatite, demonstrating the composite cement’s potential ability to promote integration with surrounding bone tissue in vivo.

### 3.5. Antibacterial Properties

To estimate the antibacterial properties of the composite cement and to verify the possible influence of the different glass grain sizes, the inhibition halo test was performed using the *S. aureus* strain. The results of the antibacterial evaluation are reported in Figure 11; as expected, the commercial samples have no antibacterial effect, instead, the composite cement containing the glass of <20 μm shows an inhibition halo of about 1 mm (black dotted line in Figure 11), which means it possesses good antibacterial effects, as reported by the standard NCCLS [34]. The composite cement C45 was not able to produce an inhibition zone; however, no bacteria proliferate under the sample, showing the composite’s ability to affect bacterial adhesion. The obtained results seem to highlight a different ability of the silver-doped glass in reducing bacterial contamination based on the size of the particles. Although both composites showed a good distribution of the glass in the polymer and especially on the cement surface, the glass particles with a smaller size have a greater specific surface area and therefore a higher ability to incorporate and release silver ions, which in turn can provide a better antibacterial effect.

## 4. Conclusions

In the present paper, composite bone cement, based on a PMMA matrix containing bioactive and antibacterial glass particles, was investigated and optimized, aiming to estimate the effect on the properties of the composite bone cement as a function of the addition of glass particles with two different grain sizes, specifically <20 µm (C20) and in the range 20–45 µm (C45), considering the need to maintain the bioactive and antibacterial behavior of the glass as well as the properties of the pristine cement as unaffected.

In detail, high-viscosity commercial bone cement was used as a polymeric matrix and the study was focused on the effect of the glass particle size on the composite cement’s physical, mechanical, and biological properties. The results obtained from the morphological analysis evidenced good dispersion of the glass particles in the commercial polymeric matrix; both composites C20 and C45 showed a good distribution of glass particles without the formation of agglomerates. Thus, both the elimination of the glass fine fraction (C20 samples) and the increase in the particle size (C45 samples) seem to avoid particle agglomeration and improve their dispersion in the polymeric matrix. Moreover, glass introduction does not negatively influence the curing parameters of commercial cement.

From the mechanical point of view, the composite cement possesses a compressive strength comparable to the commercial formulation and satisfies the ISO requirement. The bending test demonstrated that the C20 samples possess the same banding strength as the Cemex^®^ Isoplastic, while the C45 samples showed a slightly lower bending strength with respect to commercial cement, and it was slightly lower than the ISO standard requirement. This feature could reasonably have a negative effect on the primary and secondary anchorage of the cemented prostheses.

All the obtained composites proved to be strongly bioactive, evidencing the formation of a thick and uniform hydroxyapatite layer on their surface after 14 days of immersion in SBF.

Finally, a slight influence of the glass grain size was observed in the antibacterial effect of the composites, since only the C20 samples were able to create an inhibition halo, while it appears that the C45 samples were able to prevent bacterial adhesion.

Merging these results together, it appears that, to achieve appropriate mechanical properties and an antibacterial effect using this glass composition, it is necessary to have particles < 20 microns, without the fine fraction. In the future, more specific tests will be performed to confirm this hypothesis and to evaluate the cytocompatibility of the proposed composites.

The above-mentioned results demonstrate that through careful optimization of the glass amount and size, the bioactive and antibacterial composite bone cement developed in this work represents a promising material for artificial joint fixing, temporary prostheses, or spinal surgery, with the primary proven advantage of its multifunctional activity and all the specific properties already listed in the introduction. The main challenges that need to be addressed for successful translation into clinical practice can be individuated in the needed procedure for silver-containing medical devices’ certification.

## Figures and Tables

**Figure 1 materials-16-04499-f001:**
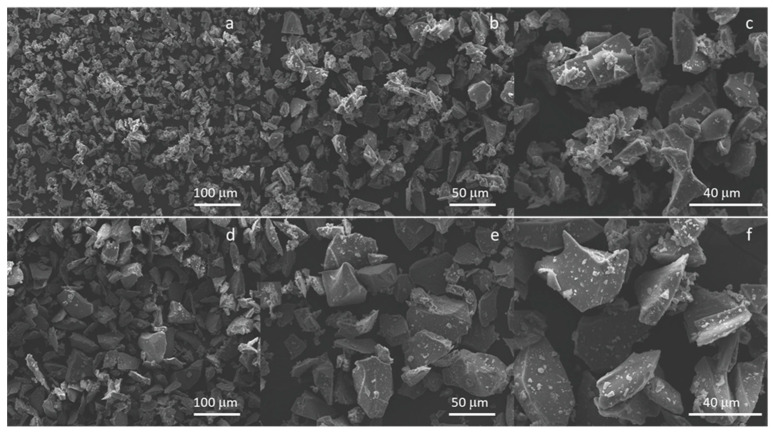
Morphology of the AgSBA2 glass powders < 20 μm (**a**–**c**) and between 20–45 μm (**d**–**f**).

**Figure 2 materials-16-04499-f002:**
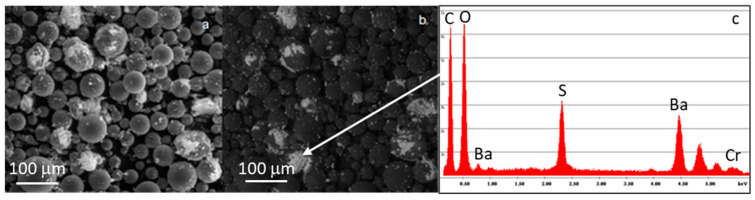
Morphology of the starting Cemex Isoplastic powders: (**a**) secondary electrons, (**b**) backscattered electrons, and (**c**) EDS analysis performed on barium sulfate particles (see arrow).

**Figure 3 materials-16-04499-f003:**
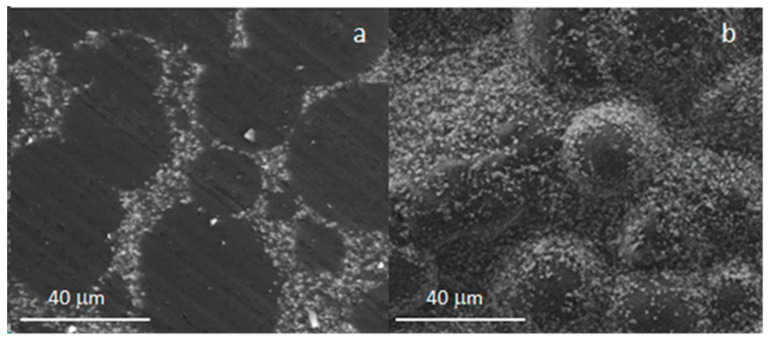
SEM analysis after hardening of the Cemex^®^ Isoplastic cross-section (**a**) and surface (**b**).

**Figure 4 materials-16-04499-f004:**
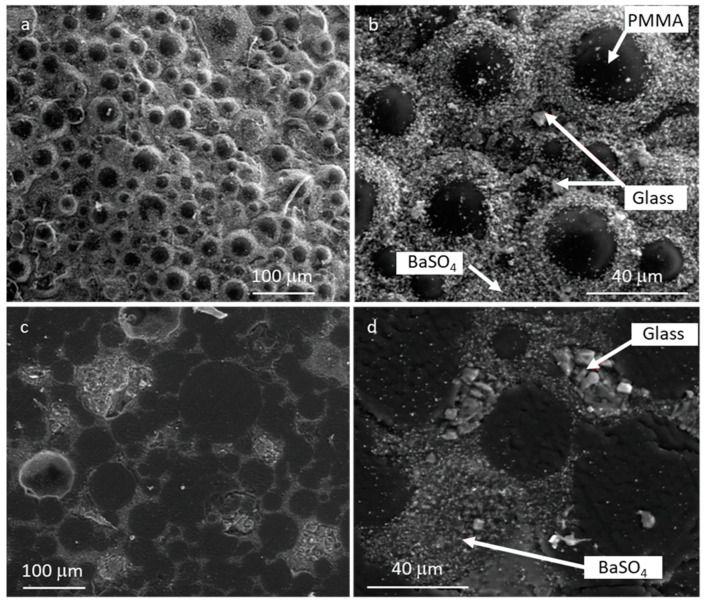
SEM analysis after hardening of the C20 surface (**a**,**b**) and the cross-section (**c**,**d**).

**Figure 5 materials-16-04499-f005:**
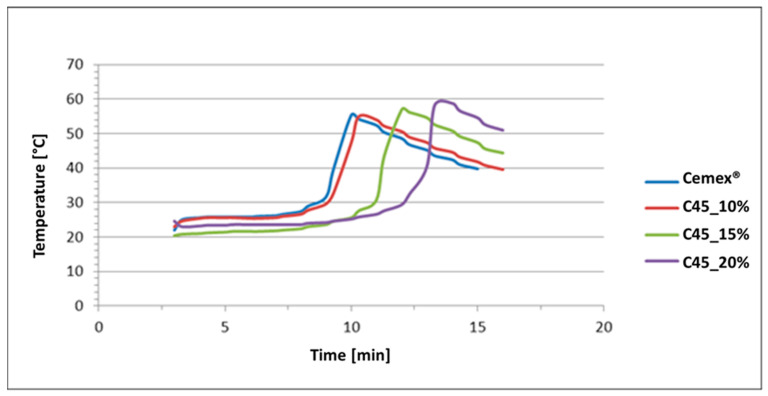
Setting time evaluation of the pure Cemex^®^ Isoplastic and composite samples.

**Figure 6 materials-16-04499-f006:**
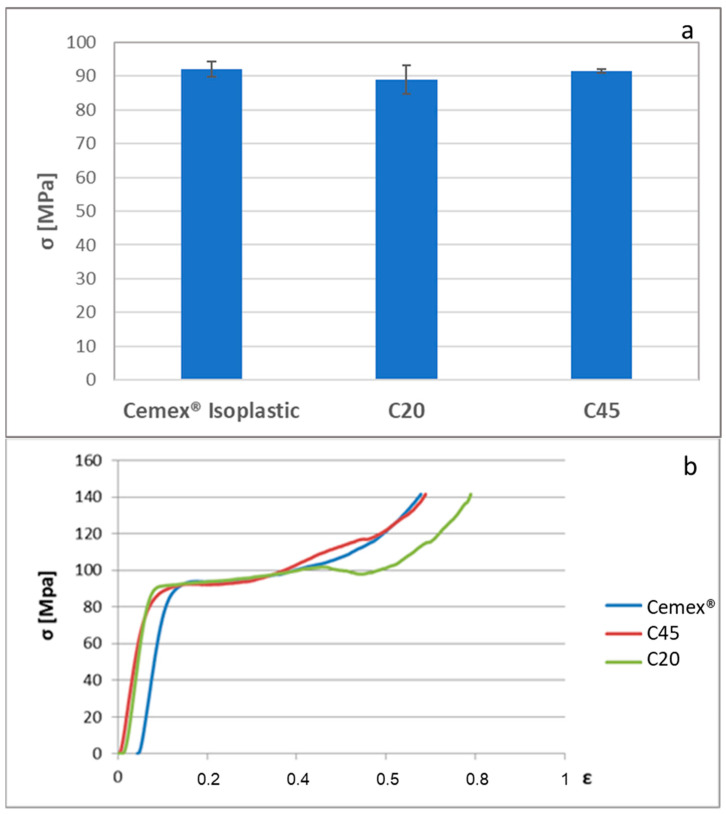
Evaluation of the compressive strength of the composite cement and Cemex^®^ Isoplastic in accordance with the ISO 5833-2002 standard. (**a**) Bars represent means and standard deviations; (**b**) example of the stress–strain curve obtained for each sample.

**Figure 7 materials-16-04499-f007:**
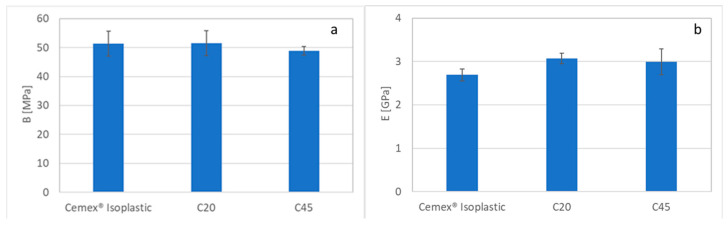
Bending strength (**a**) and modulus (**b**) evaluation of the Cemex^®^ Isoplastic and composite cement.

**Figure 8 materials-16-04499-f008:**
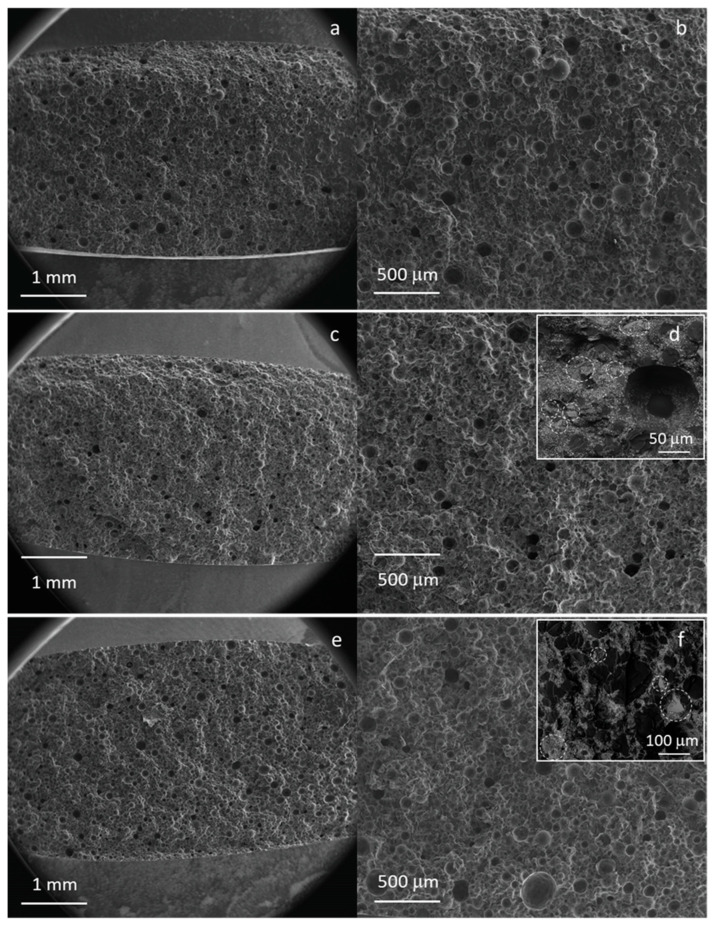
FESEM analysis of the surface fracture of the Cemex^®^ Isoplastic (**a**,**b**), C20 (**c**,**d**), and C45 (**e**,**f**). Inset: high magnification of composites; circles evidence the glass particles.

**Figure 9 materials-16-04499-f009:**
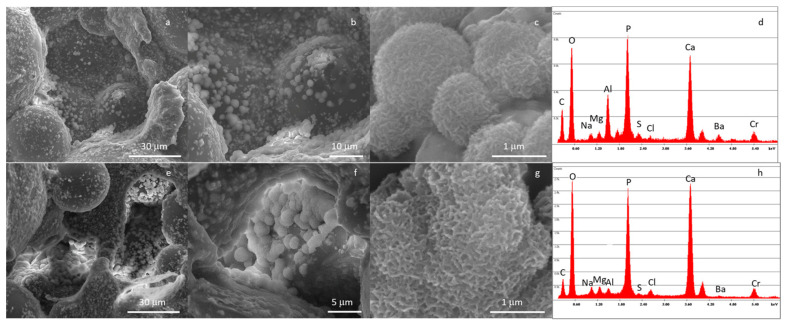
FESEM micrographs and EDS analysis of C20 (**a**–**d**) and C45 (**e**–**h**) immersed in SBF solution for up to 14 days.

**Figure 10 materials-16-04499-f010:**
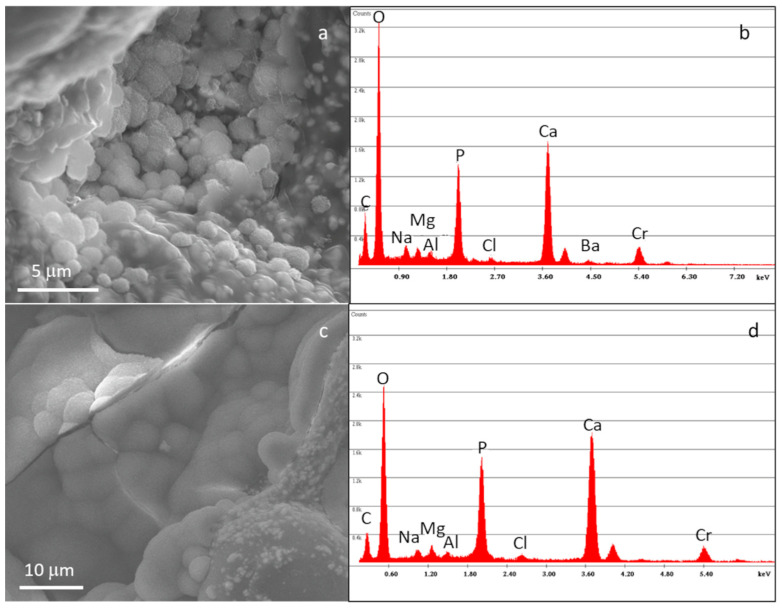
FESEM micrographs and EDS analysis of C20 (**a**,**b**) and C45 (**c**,**d**) immersed in SBF solution for up to 28 days.

**Figure 11 materials-16-04499-f011:**
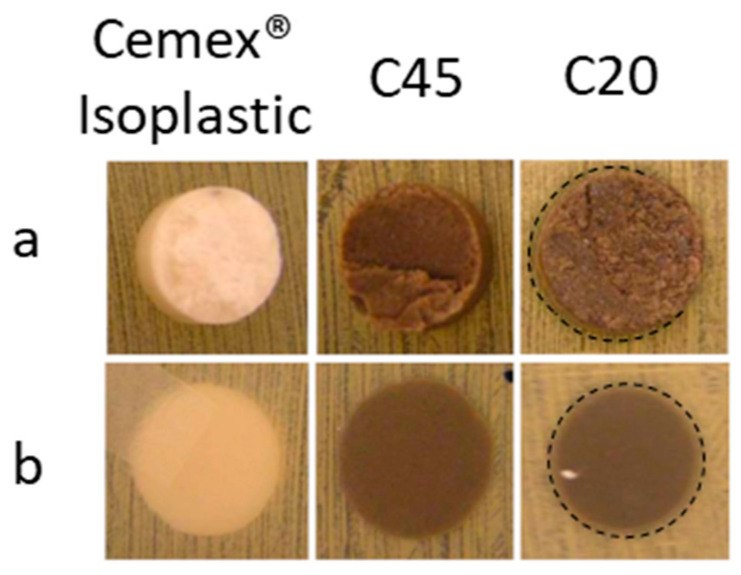
Inhibition halo of the Cemex^®^ Isoplastic, C45, and C20 samples after 24 h of incubation: (**a**) front of the plate and (**b**) back of the plate. The inhibition halo is evidenced by a black dotted line.

**Table 1 materials-16-04499-t001:** Curing parameters of the Cemex^®^ Isoplastic and the C45 composite bone cement with different % of glass particles.

	T_max_ (°C)	t_max_ (min)	T_set_ (°C)	t_set_ (min)
Cemex^®^ Isoplastic	58	9.3	40	8.3
C45_10%	51	10	37	9.3
C45_15%	57	12	39	11.3
C45_20%	59	12.3	40	12

## Data Availability

The data presented in this study are available on request from the corresponding author. The data are not publicly available due to the absence of a specific repository.

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
