# Peer review of "Physical, Mechanical, and Biological Properties of PMMA-Based Composite Bone Cement Containing Silver-Doped Bioactive and Antibacterial Glass Particles with Different Particles Sizes"

_materials, 2023, doi:10.3390/ma16134499_

Round 1

Reviewer 1 Report

The authors have investigated “Physical, mechanical and biological properties of PMMA-based composite bone cements containing silver-doped bioactive and antibacterial glass particles with different particles size”. The paper remains to the point, avoids excessive jargon, and is palatable to unfamiliar readers. The paper is also illustrated with figures that capture the topic. Also, the use of a high viscosity commercial bone cement as the polymeric matrix provides a practical approach for testing the glass particle addition. Moreover, the morphological analysis demonstrating good dispersion of glass particles and absence of agglomerates in both C20 and C45 composites strengthens the reliability of the experimental setup. In my opinion, what could improve the paper further is summarized below:

While the study investigates the effect of two different glass particle sizes, could the authors provide insights into the rationale behind choosing these specific sizes? Are there any plans to explore a wider range of particle sizes or glass amounts to optimize the balance between mechanical strength, antibacterial properties, and bioactivity?

The study mentions a slight reduction in bending strength for the composites containing larger glass particles (C45) compared to the commercial cement and ISO requirements. Can the authors discuss the potential clinical implications of this reduction? How does it affect the overall mechanical integrity and stability of the bone cement? Are there any potential strategies to mitigate this reduction while maintaining antibacterial properties?

The findings suggest that the composites with larger glass particles (C45) do not produce an inhibition halo towards S. aureus strain. Can the authors provide more insights into the potential reasons behind this observation? Are there any hypotheses or possible explanations for the lack of antibacterial effect in C45 samples?

In the conclusion, the potential use of the composite cement in artificial joints, temporary prostheses, or spinal surgery is mentioned. Can the authors further elaborate on the potential advantages, clinical feasibility, and practical implementation of the developed multifunctional device in these specific applications? Are there any challenges or limitations that need to be addressed for successful translation into clinical practice?

Author Response

The authors have investigated “Physical, mechanical and biological properties of PMMA-based composite bone cements containing silver-doped bioactive and antibacterial glass particles with different particles size”. The paper remains to the point, avoids excessive jargon, and is palatable to unfamiliar readers. The paper is also illustrated with figures that capture the topic. Also, the use of a high viscosity commercial bone cement as the polymeric matrix provides a practical approach for testing the glass particle addition. Moreover, the morphological analysis demonstrating good dispersion of glass particles and absence of agglomerates in both C20 and C45 composites strengthens the reliability of the experimental setup. In my opinion, what could improve the paper further is summarized below:

-             While the study investigates the effect of two different glass particle sizes, could the authors provide insights into the rationale behind choosing these specific sizes? Are there any plans to explore a wider range of particle sizes or glass amounts to optimize the balance between mechanical strength, antibacterial properties, and bioactivity?

Answer: in this paper we aimed to compare composites bone cements characterized by fine and coarse glass particle size. The two ranges have been chosen on the basis of previous experience [30].

The specific sizes have been selected considering that, as already reported in the original manuscript, it was demonstrated that the presence of very fine particles fraction can produce a negative effect on the mechanical properties [30,32]. For this reason, as detailed in the manuscript, the fraction < 20 µm was prepared using a mortar that avoided the formation of very fine particles (< 5 µm).

On the other hand, in a previous work, the authors evidenced, with different polymeric matrices, a slight improvement of mechanical strength using larger glass particles [30]. To verify the effective influence of a slightly larger glass powders size and to better evaluate whether the mechanical properties are more influenced by the size of the particles or by the presence of the fine fraction, they also included the 20-45 µm size in the present work. As also noticed by the reviewer, in the present manuscript it has been observed that the composites containing larger glass particles (C45) show a slight reduction in bending strength if compared to the commercial cement, composite with particles <20 micron (without fine fraction) and ISO requirements. Moreover, the antibacterial effect decreases by increasing the particles size. Thus, in our opinion, a further balance between mechanical strength, antibacterial properties, and bioactivity, if needed, cannot be done exploring a wider range of particle size, but, for example, by a further optimization of the glass composition and silver concentration. The manuscript has been improved to better explain this point.

-             The study mentions a slight reduction in bending strength for the composites containing larger glass particles (C45) compared to the commercial cement and ISO requirements. Can the authors discuss the potential clinical implications of this reduction? How does it affect the overall mechanical integrity and stability of the bone cement? Are there any potential strategies to mitigate this reduction while maintaining antibacterial properties?

Answer: the authors tried to increase the glass particle size to avoid the presence of fine fraction that can lead to the formation of aggregates and significantly reduce the mechanical properties. However, as demonstrated by the obtained data, even by increasing the size, the values of bending results are lower, even if slightly, than the limit required by the ISO standards. This feature could reasonably have a negative effect on the primary and secondary anchorage of the cemented prostheses.

To reach appropriate mechanical properties and an antibacterial effect using this glass composition, it is necessary to have particles < 20 microns, without the fine fraction.

The discussion and conclusions have been improved as requested.

-             The findings suggest that the composites with larger glass particles (C45) do not produce an inhibition halo towards S. aureus strain. Can the authors provide more insights into the potential reasons behind this observation? Are there any hypotheses or possible explanations for the lack of antibacterial effect in C45 samples?

Answer: as introduced in the manuscript, the greater specific surface area of the powders with smaller size allows both to introduce a greater amount of Ag during the ion exchange process, and therefore to release a higher amount of silver ions during the antibacterial test, producing the inhibition zone.

-             In the conclusion, the potential use of the composite cement in artificial joints, temporary prostheses, or spinal surgery is mentioned. Can the authors further elaborate on the potential advantages, clinical feasibility, and practical implementation of the developed multifunctional device in these specific applications? Are there any challenges or limitations that need to be addressed for successful translation into clinical practice?

Answer: thank you for this comment. The primary proven advantage of its multifunctional activity, and all the specific properties already listed in the introduction section. The main challenges that need to be addressed for successful translation into clinical practice can be individuated in the needed procedure for silver-containing medical devices certification. The conclusions have been improved as requested.

Reviewer 2 Report

The composite materials compose of silver-doped glass microparticle in a poly (methyl methacrylate) (PMMA) matrix is presented in the manuscript. A different particle sizes of the synthesized bioactive glass microparticle in composite materials were studied by morphology, mechanical and biological properties. However, in the results is found the different glass microparticle size does not affect the composites bioactivity and mechanical properties and the antibacterial property has not shown a significant improving. The bioactivity results showed that it promoted of in vitro grown hydroxyapatite. The obtained results demonstrate that, by the glass microparticle size, a multifunctional material for the artificial joints fixing, temporary bone can be obtained. In this work is a significant contribution to the field and is suitable for publication. But some part of a manuscript must be clarified and detailed.

(1) Authors claimed that the different of bioactive glass particle was used to study. Why you select the size of microparticle in both <20 um and 20-45 um? Two size is very close in term of scale. Instead of the size should be less than a micron, micron and more than a hundred and so on, to compare in the study. Please, give details.

(2) In the introduction, line 62-63; "but the antibacterial properties of these formulations, as well as the bone cement handling and its mechanical properties still need optimization [24]." The ref. [24] is patents that related regarding “the Composite Bone Cements with a PMMA Matrix, Containing Bioactive Antibacterial Glasses or Glass-Ceramics”. Please, declare the conflict of interest and freedom to operate. Can the manuscript be published?

(3) In the introduction, line 105-109; “in the present paper a high viscosity commercial bone cement was used as polymeric matrix and the study was focused on the effect of glass particle size on the composite cement mechanical properties, in particular bending strength, without altering the bioactive and antibacterial behavior induced by the dispersed glass particles, as well as the setting properties of pristine cement.” How is this different from the work of Enrica Verné etc. study in reference at [30]? What is the contribution of this study compare with Enrica Verné etc.work? Please, give details.

(4) In Materials and Methods, line 117, 139, 175; “two different grain sizes, i.e. < 20 um and 20÷45 μm.” What is “20÷45 μm”? Please, collects in the manuscript. 

(5) In Materials and Methods, line 114; What is the novelty in term of a novel material? Please, details in the context of manuscript.

(6) In Results and Discussion, 3.1, line 231; “about 10 and about 80 m together” Please, check the symbol “80 …m”. 

(7) In Results and Discussion, 3.2, line 253-257; The figure 5 and Table 1 are showed the effect of curing behavior of Cemex and C45 in various concentration. How about the result of C20 in various concentration? Could you rewrite of graph that can be compare in Cemex, C20 and C45? Please, more details regarding the size of C20 and C45 to the curing behavior.

(8) In Results and Discussion, 3.2, line 264; “The highest polymerization temperatures of composites do not differ from the Tmax of Cemex® Isoplastic” in the word of “Tmax”, please recheck the symbol.

(9) In Results and Discussion, 3.3.1; In the figure 6, the compression test was determined by ISO5833-2002. The results are found that Cemex, C20 and C45, were shown not significant differences in compression strength. How the effect of concentration (10%, 15%, 20% with) of C20 and C45? Because the authors presented the effect of concentration of C45 in the figure 5. Please, give details.

(10) In Results and Discussion, 3.3.2; In the figure 7, the bending test was determined by ISO standard. How the effect of concentration (10%, 15%, 20% with) of C20 and C45? Because the authors presented the effect of concentration of C45 in the figure 5. Please, give details.

(11) In the figure 11, the picture of inhibition halo of Cemex, C20 and C45 is not clear. Could you improve the quality of the image? Please, descript more regarding why the Ag-SBA microparticle is not active. Is it regarding the concentration of Ag-SBA microparticle in composites? 

(12) In Conclusion, line 371; “the range 20÷45 μm (C45)”, please, recheck the symbol “20÷45 μm”

(13) The format of reference should be improve, please recheck and collect by the format of the journal. 

Minor editing of English language required. Please, recheck the abbreviations, symbols and units.

Author Response

The composite materials compose of silver-doped glass microparticle in a poly (methyl methacrylate) (PMMA) matrix is presented in the manuscript. A different particle sizes of the synthesized bioactive glass microparticle in composite materials were studied by morphology, mechanical and biological properties. However, in the results is found the different glass microparticle size does not affect the composites bioactivity and mechanical properties and the antibacterial property has not shown a significant improving. The bioactivity results showed that it promoted of in vitro grown hydroxyapatite. The obtained results demonstrate that, by the glass microparticle size, a multifunctional material for the artificial joints fixing, temporary bone can be obtained. In this work is a significant contribution to the field and is suitable for publication. But some part of a manuscript must be clarified and detailed.

(1)             Authors claimed that the different of bioactive glass particle was used to study. Why you select the size of microparticle in both <20 um and 20-45 um? Two size is very close in term of scale. Instead of the size should be less than a micron, micron and more than a hundred and so on, to compare in the study. Please, give details.

Answer: in this paper we aimed to compare composites bone cements characterized by fine (< 20 µm) and coarse (20-45) glass particle size. The two ranges have been selected on the basis of previous experience [30].

The specific sizes have been chosen considering that, as already reported in the original manuscript, it was demonstrated that the presence of very fine particles fraction can produce a negative effect on the mechanical properties [30, 32]. For this reason, as detailed in the manuscript, the fraction < 20 µm was prepared using a mortar that avoided the formation of very fine particles (< 5 µm).

On the other hand, in a previous work, the authors evidenced, with different polymeric matrices, a slight improvement of mechanical strength using larger glass particles [30]. To verify the effective influence of a slightly larger glass powders size and to better evaluate whether the mechanical properties are more influenced by the size of the particles or by the presence of the fine fraction, they also included the 20-45 µm size in the present work. The obtained results revealed that the composites containing larger glass particles (C45) show a slight reduction in bending strength if compared to the commercial cement, composite with particles <20 micron (C20, without fine fraction) and ISO requirements. Moreover, the antibacterial effect decreases by increasing the particles size. Thus, in our opinion, a further balance between mechanical strength, antibacterial properties, and bioactivity, if needed, cannot be done exploring a wider range of particle size, but, for example, by a further optimization of the glass composition and silver concentration. The manuscript has been improved to better explain this point.

(2) In the introduction, line 62-63; "but the antibacterial properties of these formulations, as well as the bone cement handling and its mechanical properties still need optimization [24]." The ref. [24] is patents that related regarding “the Composite Bone Cements with a PMMA Matrix, Containing Bioactive Antibacterial Glasses or Glass-Ceramics”. Please, declare the conflict of interest and freedom to operate. Can the manuscript be published?

Answer: thank you for this comment. The patent reported in [24] is co-authored by the authors of the manuscript (M.M. and E.V.); the patent itself has been granted in Europe, but it is not licensed and it is no longer active (it was recently abandoned). So, there are not conflicts of interest.

(3) In the introduction, line 105-109; “in the present paper a high viscosity commercial bone cement was used as polymeric matrix and the study was focused on the effect of glass particle size on the composite cement mechanical properties, in particular bending strength, without altering the bioactive and antibacterial behavior induced by the dispersed glass particles, as well as the setting properties of pristine cement.” How is this different from the work of Enrica Verné etc. study in reference at [30]? What is the contribution of this study compare with Enrica Verné etc. work? Please, give details.

Answer: the composite bone cement developed in [30] was based on a different polymeric formulation, i.e. a commercial cement (G1TM) with standard viscosity, and the research reported in [30] was focused both on the optimization of the glass dispersion in the polymeric matrix and on the investigation of the effect of the glass preparation, amount, grain-size and humidity on the final properties of the composite cement. The composite bone cement developed in the present paper is made with a different high viscosity commercial bone cement (Cemex® Isoplastic) as polymeric matrix and, as reported in the manuscript, the glass particle size distribution was further optimized avoiding the fraction < 5 mm. The composite cement properties were then investigated, including the setting times, never reported before for these formulations.

(4) In Materials and Methods, line 117, 139, 175; “two different grain sizes, i.e. < 20 um and 20÷45 μm.” What is “20÷45 μm”? Please, collects in the manuscript.

Answer: the manuscript has been improved, as requested, with a better specification of the grain sizes.

(5) In Materials and Methods, line 114; What is the novelty in term of a novel material? Please, details in the context of manuscript.

Answer: the novelty is not the glass composition, that has been designed and optimized by the authors in previous works [29,30], but in the composite formulation in terms of polymeric matrix viscosity, and the glass particle size distribution avoiding the fraction < 5 m. The manuscript has been improved, as requested.

(6) In Results and Discussion, 3.1, line 231; “about 10 and about 80 m together” Please, check the symbol “80 …m”.

Answer: the manuscript has been revised, as requested, in the submitted version the symbol is mm.

(7) In Results and Discussion, 3.2, line 253-257; The figure 5 and Table 1 are showed the effect of curing behavior of Cemex and C45 in various concentration. How about the result of C20 in various concentration? Could you rewrite of graph that can be compare in Cemex, C20 and C45? Please, more details regarding the size of C20 and C45 to the curing behavior.

Answer: we did not investigate the curing parameters for C20, because the effects on the setting time are mainly due to the amount of filler (bioactive glass in this case) introduced into the composite, which can slow down the polymerization reaction or partially prevent it. Since we expected a more evident effect with the bigger particle size only C45 samples have been used to underline eventual differences in the various glass particle concentrations. This aspect was specified in the text.

(8) In Results and Discussion, 3.2, line 264; “The highest polymerization temperatures of composites do not differ from the Tmax of Cemex® Isoplastic” in the word of “Tmax”, please recheck the symbol.

Answer: the symbol has been corrected, as requested.

(9) In Results and Discussion, 3.3.1; In the figure 6, the compression test was determined by ISO5833-2002. The results are found that Cemex, C20 and C45, were shown not significant differences in compression strength. How the effect of concentration (10%, 15%, 20% with) of C20 and C45? Because the authors presented the effect of concentration of C45 in the figure 5. Please, give details.

Answer: The effect of glass amount on the mechanical properties of composite cements (in particular on compression strength) has been deeply investigated by the authors in previous works [27, 30], evidencing a significant reduction of the compression strength by increasing the glass amount. This paper is more focused on the investigation of the glass particles size on cement’s properties.

(10) In Results and Discussion, 3.3.2; In the figure 7, the bending test was determined by ISO standard. How the effect of concentration (10%, 15%, 20% with) of C20 and C45? Because the authors presented the effect of concentration of C45 in the figure 5. Please, give details.

Answer: also in this case, since the effect of the concentration of glass particles on mechanical properties was investigated in previous work, the authors focused the attention on the effect of different glass particles sizes.

(11) In the figure 11, the picture of inhibition halo of Cemex, C20 and C45 is not clear. Could you improve the quality of the image? Please, descript more regarding why the Ag-SBA microparticle is not active. Is it regarding the concentration of Ag-SBA microparticle in composites?

Answer: the figure has been improved evidencing the inhibition halo. As reported in the manuscript, we hypothesized that the glass particles with a smaller size have a greater specific surface and therefore a higher ability to incorporate and release silver ions, that in turn can provide a better antibacterial effect. The manuscript has been improved to underline this point.

 (12) In Conclusion, line 371; “the range 20÷45 μm (C45)”, please, recheck the symbol “20÷45 μm”

Answer: the manuscript has been improved, as requested.

(13) The format of reference should be improved, please recheck and collect by the format of the journal.

Answer: we rechecked, as requested, and we guess that the format of reference is correct.

Comments on the Quality of English Language. Minor editing of English language required. Please, recheck the abbreviations, symbols and units.

Reviewer 3 Report

I have to notice first that the authors have already published a version of this manuscript, i.e. preprint, not reviewed by any journal: https://www.researchsquare.com/article/rs-2071896/v1.  I hope that this is OK with Materials journal policy.

This is an interesting topic and the authors have performed many tests to show performances of this new bone cement. I have some remarks and questions given in comments attached to the manuscript pdf file.

English language and grammar should be checked carefully, I marked yellow some mistakes I have noticed, and added some comment in the manuscript pdf file.

Author Response

I have to notice first that the authors have already published a version of this manuscript, i.e. preprint, not reviewed by any journal: https://www.researchsquare.com/article/rs-2071896/v1.  I hope that this is OK with Materials journal policy.

Answer: the cited preprint, probably refers to a previous submission that was not reviewed nor accepted for the publication, but it is licensed under a CC BY 4.0 License, that reports that the authors are “free to: Share — copy and redistribute the material in any medium or format Adapt — remix, transform, and build upon the material for any purpose, even commercially”. Moreover, in the present submission the manuscript has been modified and improved according to the reviewer’s comments.

-             This is an interesting topic and the authors have performed many tests to show performances of this new bone cement. I have some remarks and questions given in comments attached to the manuscript pdf file.

-             Comments on the Quality of English Language: English language and grammar should be checked carefully, I marked yellow some mistakes I have noticed, and added some comment in the manuscript pdf file.

Answer: thank you very much for this comment, the manuscript has been improved, as requested. The revised version contains all the suggested corrections.

Round 2

Reviewer 2 Report

The composite materials compose of silver-doped glass microparticle in a poly (methyl methacrylate) (PMMA) matrix is presented in the manuscript. A different particle sizes of the synthesized bioactive glass microparticle in composite materials were studied by morphology, mechanical and biological properties. However, the author has commented and revised the manuscript in relevant parts. In this work is a significant contribution to the field and is suitable for publication.